# Preparation of ETS-10 Microporous Phase Pellets with Color Change Properties

**DOI:** 10.3390/gels5030042

**Published:** 2019-09-02

**Authors:** Pierantonio De Luca, Carmelo Mastroianni, Carlo Siciliano, Janos B. Nagy, Anastasia Macario

**Affiliations:** 1Department of Ingegneria Meccanica, Energetica e Gestionale, University of Calabria, Arcavacata di Rende, 87036 Rende (Cs), Italy; 2Department of Ingegneria per l’Ambiente ed il Territorio ed Ingegneria Chimica, University of Calabria, Arcavacata di Rende, 87036 Rende (Cs), Italy; 3Dipartimento di Farmacia e Scienze della Salute e della Nutrizione, Università della Calabria, I-87036 Arcavacata di Rende (CS), Italy

**Keywords:** ETS-10, catalysis, pH responsive materials, microporous materials, pellets

## Abstract

The main scope of the present work is to synthesize pH-responsive Engelhard titanium silicate (ETS)-10 phase crystalline pellets through the smart modification of a synthetic process which was previously applied to the preparation of other phases. The original preparative method, which envisages the use of the same initial synthesis as a binder for the preparation of pellets, was modified by adding an appropriate pH indicator to a number of systems subject to this investigation. It should be noted that the modified process was never before used to give access to pH-responsive ETS-10 phase pellets, and it is disclosed here for the first time. The study started from the definition of the best experimental conditions, which were optimized by analyzing the effects of temperature and system composition. The addition of the pH indicator did not alter the physicochemical characteristics and reactivity of the system. The pH-responsive ETS-10 phase crystalline pellets were characterized by an adequate mechanical strength and by a high capability to change color. The latter aspect can be particularly useful when this material is used in catalytic processes whose performance is strictly dependent on the pH value. The amount of gel used, as well as the working temperature, were the main critical parameters to be controlled during the preparation of pH-responsive ETS-10 phase crystalline pellets. The pellets were fully characterized by X-ray diffraction in order to investigate and identify the possible phases, and by using a hardness tester to measure the compressive strength. Finally, toning tests were performed.

## 1. Introduction

The environmental problem, now more and more incumbent, is being tackled in various fields of research, such as that of eco-sustainable materials with low environmental impact [1,2,3,4,5,6,7,8], but also through new materials that can be used for the removal and photodegradation of environmental pollutants [9,10]. Microporous materials always aroused particular interest [11,12,13,14].

The ETS-10 phase is a microporous material that was synthesized for the first time by Kuznicki [15] with particularly important applications in the environmental field. It belongs to the Engelhard titanium silicate (ETS) family and is characterized by a structure with a wide pore opening of about 0.8 nm [16,17,18]. In recent decades, many studies paid particular attention to the synthesis of this material which is generally obtained through hydrothermal synthesis [19,20,21,22,23,24].

Data from the literature report that hetero atoms were inserted into the structure of these materials to modify their properties [25,26]. This material can be used in many sectors thanks to its particular peculiarities. It has properties of ion exchange [27,28,29], adsorption [30,31,32,33,34,35], and also photocatalytic properties [36,37,38,39,40,41]. The coexistence of these properties makes these materials particularly and potentially interesting thanks to the possibility of creating a perfect synergy between them, which can be applied advantageously for the removal and degradation of common environmental pollutants such as polycyclic aromatic hydrocarbons, cycloalkanes, benzene, toluene, ethylbenzene, etc. [42,43,44,45,46].

Generally, the ETS-10 phase is synthesized in powder form, but in the last few years a particular interest was directed towards the direct synthesis of self-bonded pellets [47] because the ETS-10 phase in the form of pellets allows for greater ease of use and recovery. Furthermore, the absence of the binder in self-bonded pellets avoids the obstruction of the porous structure with a consequent decrease in the accessibility of the pores and therefore a reduction of their properties. More recent studies report the synthesis of self-linked ETS-10 phase pellets with the use of carbon nanotubes [48]. 

The aforementioned reasons, and the considerable importance of the ETS-10 phase, justify the particular interest in the possibility of using simple, advantageous, and cost-effective methods for the preparation of such material in a form which can present a greater possibility of recovery after its use, with a total environment preservation. Therefore, the scope of this investigation is to synthesize ETS-10 phase crystalline pellets, which are able to assume different colorations, depending on the pH value of the environment where these materials are working. The color change of the ETS-10 phase crystalline pellets might be a characteristic of particular importance when this material is applied for catalytic processes whose performance is thought to be highly dependent on the pH value in which the same process takes place. The pellet form, and the desired physico-chemical characteristics of the pH-responsive ETS-10 phase crystalline, could make this material a valid and ideal candidate as a simple pH-responsive sensor for the control of the purposes and results of catalytic processes. To our knowledge there are no studies on pH-responsive ETS-10 phase crystalline pellets, indicating the need for the investigation of the most appropriate synthetic conditions in order to obtain this material with the characteristics discussed above. We started our work searching for an appropriate modification of an already published preparative methodology, which was successfully applied for the production of other phases [49].

In particular, the new method uses the crystalline phase and its own initial gel, which after a minimal pre-treatment is employed as a binder for the preparation of pellets. This method is used for the first time as a tool for the preparation of ETS-10 phase crystalline pellets. A further modification of the original method is obtained by adding an additive which can act as a pH indicator. The effects of adding this unusual component to the ETS-10 phase precursor system, known to be particularly sensitive to any modification, are carefully investigated. 

## 2. Results and Discussion

Figure 1 shows the images spectra of the pellets as a function of the ETS-10_crystalline_/xerogel ratio and of the cooking or treatment conditions. The addition of the Congo Red did not lead to a change in the pH of the system that remained around 10.5.

Figure 1 shows that not all prepared pellets retained their shape, particularly the pellets with ETS-10_crystalline_/xerogel = 9 treated at 190 °C by hydrothermal synthesis. Most of the pellets had a reddish color typical of the Congo Red indicator, with the exception of pellets obtained with ETS-10_crystalline_/xerogel ratios equal to 3, 9, and treated by hydrothermal synthesis at 190 °C.

Figure 2 shows the XRD spectra relative to the pellets obtained at ETS-10_crystalline_/xerogel = 3 and 9 ratio and treated at 25, 100, and at 190 °C, it is mainly evident that the addition of Congo Red does not disturb the reactivity of the systems that evolves towards the formation of the ETS-10. In addition to the characteristic peaks of the ETS-10 phase, there are also other peaks attributable to the xerogel that remained in amorphous form and to minor phase parasites such as titanates.

All pellets prepared with ETS-10_crystalline_/xerogel = 1 ratio showed a low crystallinity, attributable to the large amount of xerogel present that would need longer times to crystallize completely.

In light of these results it is possible to explain the loss of color of the pellets obtained with ETS-10_crystalline_/ xerogel = 3 and 9 by hydrothermal synthesis at 190 °C. This can be attributed both to the synthesis process and to the degree of crystallinity of the pellets. The hydrothermal process, due to its intrinsic characteristics, takes place in an environment saturated with steam, where the release of the indicator from pellets during the cooking is favored. 

Furthermore, this release is amplified when, owing to an increase in crystallinity, the presence of xerogel decreases, to which the ability to retain the indicator is delegated.

In the following, Figure 3 shows the values of mechanical resistance of the pellets subjected to tests of mechanical compressive strength.

The data obtained show a loss of mechanical strength especially for pellets prepared by hydrothermal synthesis. Furthermore, it is possible to underline that the lowering of the resistances is mainly recorded in the pellets characterized by a high crystallinity and therefore a lower presence of xerogel.

Resistance values vary depending on the two reference variables: ETS-10_crystalline_/xerogel ratio and the cooking or treatment conditions. The highest values obtained at *T* = 100 °C were equal to 130.42 and 124.54 N respectively, with the ETS-10_crystalline_/xerogel ratio equal to 1 and 3.

The pellets prepared with ETS-10_crystalline_/xerogel = 3 at *T* = 100 °C proved to be interesting because they showed good crystallinity and good mechanical strength. This system probably identifies the best synthesis conditions to prepare the pellets with Congo Red.

The pellets obtained dipped in water for 2 h at room temperature were observed to preserve their shape, their mechanical characteristics, and their color, thus allowing to exclude the loss of the indicator in the water and confirming that it is well anchored inside of pellets.

The pellets prepared with Congo Red were then subjected to toning tests to ascertain their ability to change color.

In particular, two solutions were prepared, one of sodium hydroxide at pH = 13 and one of hydrochloric acid at pH = 1. Only those pellets which were obtained intact and resistant were submitted to this test. The test was carried out by immersing the pellets in the basic NaOH solution for 5 min and after observing the color change were washed with distilled water and immersed again in the HCl acid solution for another 5 min. The pellets all showed the ability to change color reversibly depending on the pH of the solution used

Figure 4 shows the images of the ETS-10 phase pellets prepared with Congo Red, obtained at ETS-10_crystalline_/xerogel = 3 ratio and treated at 100 °C, after 5 min of immersion in basic solution (Figure 4a), and in an acid solution (Figure 4b). This sample proved to be the best compromise between crystallinity and mechanical properties.

## 3. Conclusions

The method of preparation of self-bonded pellets is used for the preparation of pH-responsive ETS-10 phase crystalline pellets. This method does not use as binder substances different from the system. It allows us to hold the chemical characteristics typical of the ETS-10 phase.

The best system identified that allows for pH-responsive ETS-10 phase crystalline pellets, combining high crystallinity and high mechanical strength, is obtained for an ETS-10_crystalline_/xerogel = 3 ratio and treatment at 100 °C.

The pellets obtained with Congo Red, undergoing toning tests, showed the ability to change color reversibly depending on the pH of the solution. Given the significant importance of the ETS-10 phase and its wide applicability, in this form and with these characteristics, it is proposed as an excellent material to be used in catalytic processes for its advantageous recovery and its valuable characteristic of a simple pH-sensor for the control of the process parameters and performance.

## 4. Materials and Methods

To synthesize the ETS-10 phase pellets a method reported in the literature was applied for other phases [49].

This involved the initial preparation of the so-called xerogel, the synthesis of the crystalline phase ETS-10 powder, and the preparation of the pellets.

In particular, an initial hydrogel was prepared by synthesis method reported by Turta et al [24] utilizing the following molar system:1.0Na_2_O-0.6KF-0.2TiO_2_-1.28HCl-1.49SiO_2_-39.5H_2_O(1)

To prepare the initial hydrogel, two systems were prepared: one acid (containing hydrochloric acid (37 wt%, Carlo Erba, Milan, Italy), titanium tetrachloride solution (50 wt%, Merck, Darmstadt, Germany), potassium fluoride solution (40 wt%, Merck, Darmstadt, Germany), distilled water, and one basic (containing sodium silicate solution (8 wt% Na_2_O, 27 wt% SiO_2_, Merck, Darmstadt, Germany) and sodium hydroxide (50 wt%, Carlo Erba, Milan, Italy). In particular, the following weight amounts of the reactants were used to prepare the hydrogel: sodium silicate solution = 33.16 g; sodium hydroxide solution = 9.23 g; potassium fluoride solution = 8.71 g; hydrochloric acid = 4.70; titanium tetrachloride solution = 7.65 g; distilled water = 32.60 g.

Subsequently the two solutions were mixed together. The hydrogel was dense and white in color (Figure 5a). The pH of the hydrogel was 10.5.

The hydrogel was dried at 100 °C for 24 h and then powdered, obtaining the xerogel (Figure 5b).

The preparation of hydrogel and xerogel were found to be fundamental for the preparation of pellets. In particular the hydrogel brought with it all the chemical components so that after the hydrothermal reaction it crystallized with the formation of the ETS-10 phase. As reported in other studies [20,24], the hydrogel system was particularly sensitive to chemical variations, in fact small alterations, for example of its pH, make the hydrogel no longer reactive to crystallize as ETS-10. The powdered xerogel presented itself as a white powder (Figure 5b), potentially it still contained all the chemical components and all the characteristics to evolve, under appropriate experimental conditions, towards the crystallization of the ETS-10 phase. During the formation of hydrogel was the dissolution of all the added components and the first forms of aggregations that would lead to the formation of crystals. Thermally drying the hydrogel to obtain the xerogel allowed the loss of the liquid component, which is basically water, but it still preserved all the conditions for its transformation towards the crystalline phase. In fact, all the primary aggregates and all the elements necessary to nourish the growth of the crystals continued to be present. Therefore, the xerogel is a particularly important component for the preparation of pellets as it can still favor the growth of crystals. Furthermore, its addition in the preparation of the pellets is advantageous because, in addition to not disturbing the chemical composition of the system, the xerogel may have binder characteristics. In fact, after adding it with water, it transforms like a mortar which, by subsequent drying or heat treatment, hardens and acts as a true binder. Therefore, the xerogel in the preparation of the ETS-10 phase pellets can have a dual function, that of forming or favoring the growth of the phase crystals ETS-10, if the experimental conditions favor it, such as a suitable heat treatment, or to act only as a binder, remaining in a semi-amorphic phase. Kostov-Kytin et al. [50] also analyzed the conditions for crystallization of titanosilicate phases, however these authors obtained pure crystals and not self-bonded pellets. Figure 6 shows the XRD-diffraction of the xerogel.

In particular, to synthesize ETS-10 crystals, the hydrogel prepared in the manner described above was used directly without drying it, and inserted in Morey-type autoclaves to subject it to hydrothermal synthesis for a period of 5 days at a temperature of 190 °C.

Below, Figure 7 shows the XRD spectrum of the obtained ETS-10 crystalline phase. The XRD patterns are in good agreement with the published ETS-10 data [22,24]. 

After preparing the xerogel and the ETS-10 crystals, we proceeded with the preparation of the pellets, preparing a mixture at different weight ratios of ETS-10_crystalline_/xerogel = 1, 3, and 9. A few milliliters of distilled water was added to obtain a more workable mixture. Finally, to prepare the pellets with the ability to change color, predetermined quantities of an indicator were added directly into the mixture before the pelletizing and baking phase. The choice of the indicator to be used fell on Congo Red and the quantities used were 0.5% compared to the total weight.

Subsequently, from this mixture, pellets were preformed (dimensions: 1.3 cm in diameter and 0.7 cm in thickness), by a press using a pressure of 400 bar. Three pellets were prepared for each system studied. 

In order to find the best preparation conditions, the pellets were subjected to three distinct types of treatment: (1) They were left at room temperature (25 °C) for a day; (2) they were placed directly in the oven at 100 °C for a day; (3) they were subjected to hydrothermal synthesis at 190 °C for three days. In the latter case, the pellets were placed in Morey-type autoclaves in which a support made of Teflon was positioned which had the purpose of maintaining the raised pellet at 5 mm from the bottom. At the end of the hydrothermal reaction, the pellets were removed from the autoclave, washed with distilled water, and dried in an oven at 100 °C for one day.

The materials were characterized by a XRD spectrophotometer (Philips PW 1730/10, Panalytical, Kassel, Germany). For the tests of mechanical strength, a hardness tester (VK200 Tablet hardness, Vanderkamp, Los Angeles, CA, USA) was used. In addition, the pellets were subjected to color change tests.

## Figures and Tables

**Figure 1 gels-05-00042-f001:**
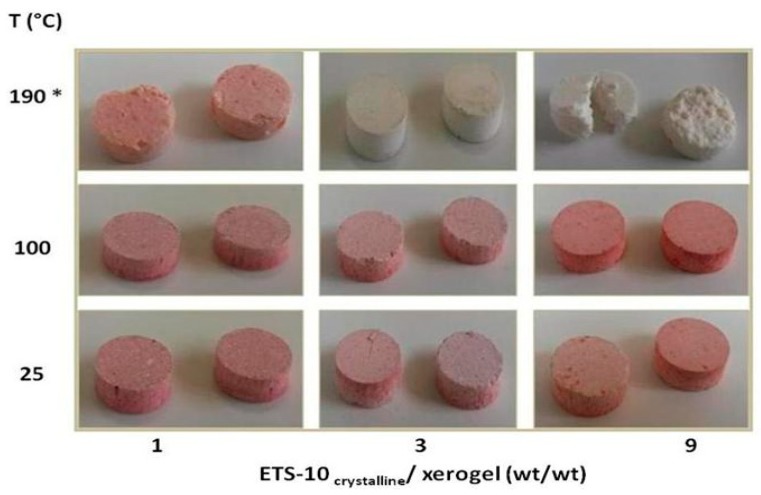
Images of the pellets as a function of the Engelhard titanium silicate (ETS)-10_crystalline_/xerogel ratio and of the cooking or treatment conditions. (*) Hydrothermal synthesis.

**Figure 2 gels-05-00042-f002:**
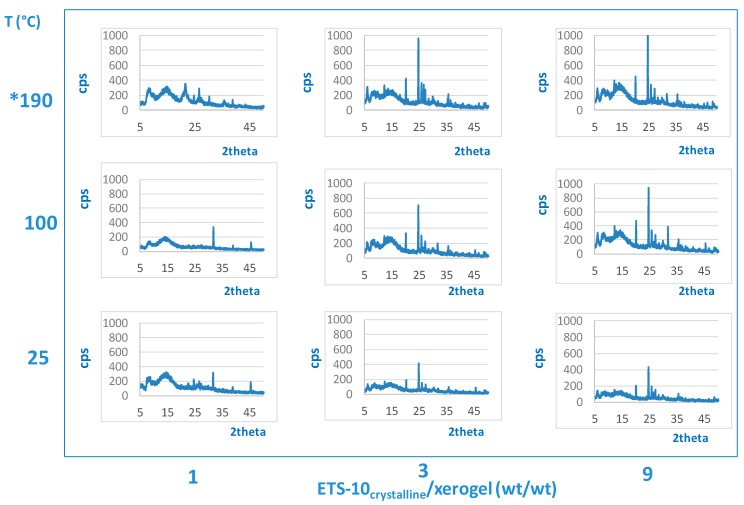
X-ray diffraction spectra of pellets a function of the ETS-10_crystalline_/xerogel ratio and of the cooking or treatment conditions. (*) Hydrothermal synthesis.

**Figure 3 gels-05-00042-f003:**
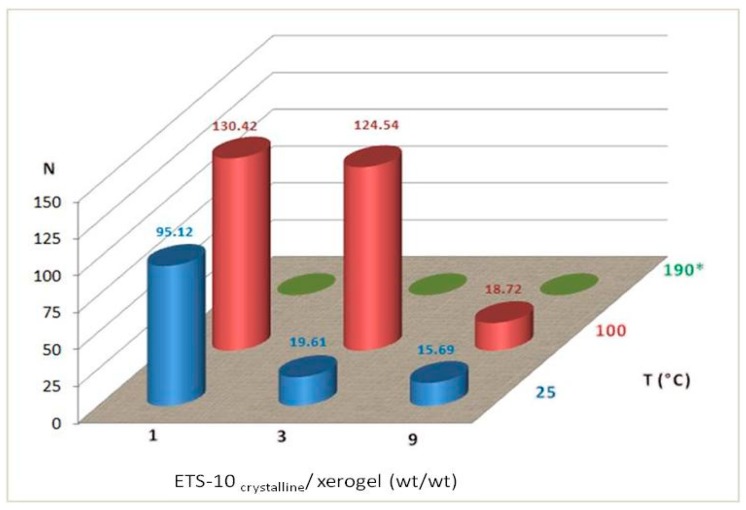
Mechanical compressive strength of the pellets as a function of the ETS-10_crystalline_/xerogel ratio and the cooking or treatment conditions. (*) Hydrothermal synthesis.

**Figure 4 gels-05-00042-f004:**
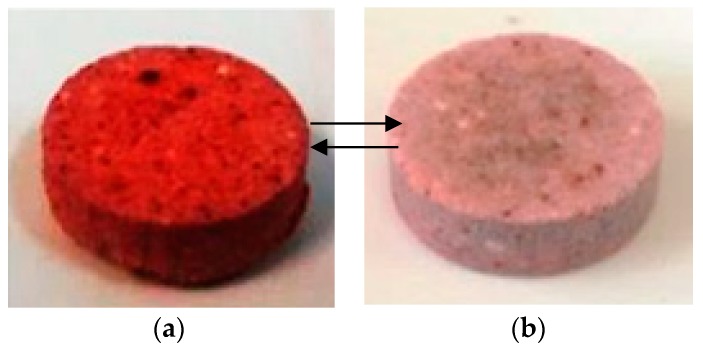
ETS-10 phase pellets with Congo Red, obtained at ETS-10_crystalline_/xerogel = 3 ratio and treated at 100 °C, after 5 min of immersion (**a**) in NaOH solution and (**b**) in HCl solution.

**Figure 5 gels-05-00042-f005:**
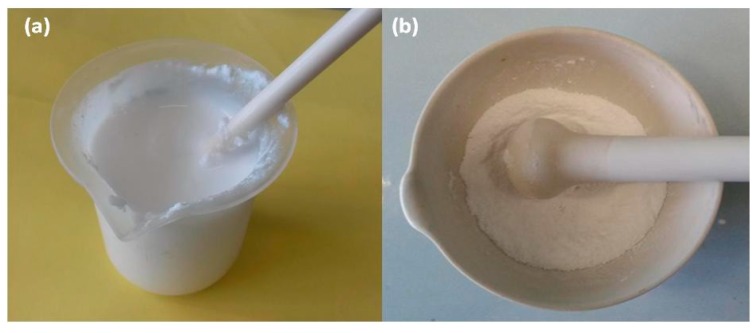
(**a**) Hydrogel; (**b**) Powdered xerogel.

**Figure 6 gels-05-00042-f006:**
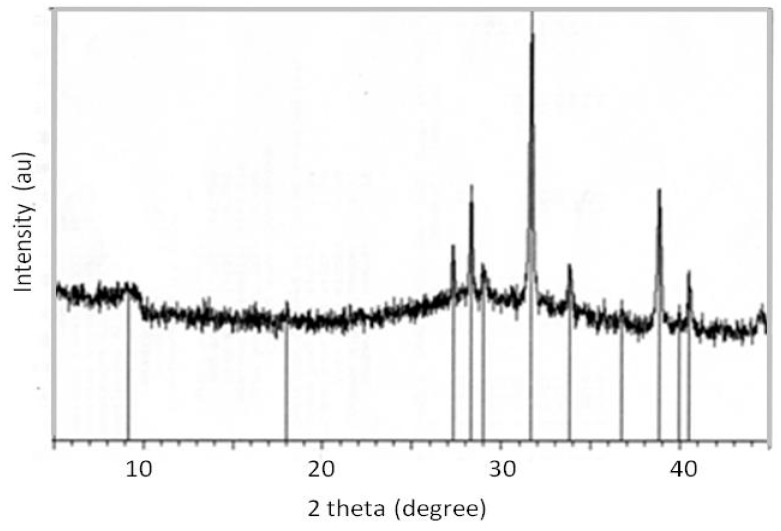
XRD spectrum of the xerogel [49].

**Figure 7 gels-05-00042-f007:**
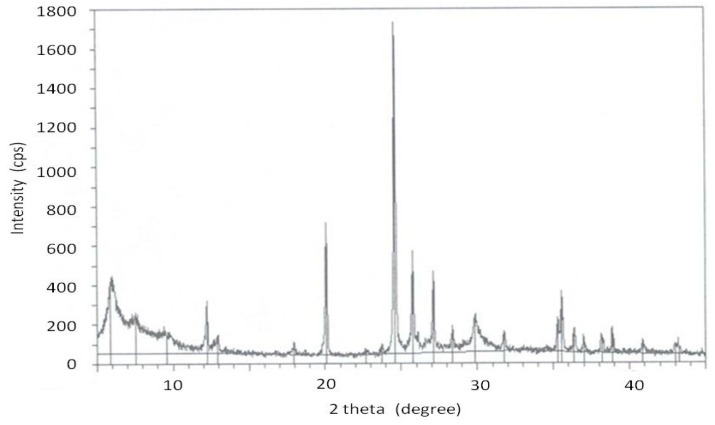
XRD spectrum of the ETS-10 phase.

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
