# Peer review of "Preparation of ETS-10 Microporous Phase Pellets with Color Change Properties"

_gels, 2019, doi:10.3390/gels5030042_

Round 1

Reviewer 1 Report

Major corrections:           Grammar, punctuation, and style must be seriously revised.

The methodology part needs revision. Synthesis details have to be omitted from the introduction and discussion. At the same time, the procedure of synthesis must be presented clearer and more detailed.

XRD patterns not properly analyzed, literature references need to attributed to peaks on patterns.

Rewrite the Conclusions.

Do you consider biotoxicity and cancerogenic properties of the pigment you have chosen? You started the paper with attitude to solve the Environmental problems, in such case toxic dye is not the best choice. Or the introduction must be focused on the catalysis that can be benefited by such an idea.

What catalytic reaction are you going to use as a test reaction? how do you choose the size of the pellet?

Minor corrections:

1.       P1, Abstract        Rewrite first sentence, it is confusing due to a dot (line 15) in the middle of the sentence.

2.       Keywords            You have mentioned in abstract catalysis as a main area of application, but say nothing about it in the Keywords. Maybe, highlighting that moment in keywords will be helpful.

3.       Introduction. Line 37.     2nd sentence is too much generic. It is advisable to make it more specific to the purpose or to avoid it at all.

a.       L. 49     Provide some examples of target environmental pollutions

b.       L. 67       To our knowledge, there are ….

c.       L. 71      other phases of Titanosilicate materials?

d.       Lines 71-78. Those are not the introduction sentences. It would be better to move them to part 2. Materials and methods

4.        Materials and methods               1st sentence need to be revised: make it simpler, consequently, punctuation will be easier.

5.       L. 83      an initial gel

6.       L. 90      Sentence need to be rewritten (For example: Thereafter, hydrogel was dried at…. and obtained xerogel was grinned using such and such tool). Work on synonyms, there is too much repetition. Working with gels you ought to know that fresh gel you obtained is a hydrogel. It is important to mention because you are drying them afterward obtaining xerogels. Please, use scientific terminology.

7.       Figure 1.a            It is not informative and, rather, confusing. It is not clear whether it is a white paste or white gel. When you are taking gel out of the beaker, are the portions of gel keeping their shape?

If you want to prove and show that it really was gel your pictures must be different. For example, the stirring stick must stand upright, not touching the side of the beaker. Another way: you incline the beaker, but gel’s surface does not move. Picture of the gel without the beaker, that holding its own shape would be more informative as well.

8.       Figure 1               it would be nice to present a picture of dried gel before it was grined as well.

9.       Lines 103 and 106             do you mean crystals or crystallites?

10.   Lines 100-106     It is good that you are trying to explain and not just providing the facts.

Conditions for crystallization the titanosilicate phases (pH, composition and temperature influence) were studied in details by V. Kostov-Kytin and colleagues (Hydrothermal synthesis of microporous titanosilicates.  Microporous and Mesoporous Materials 105(3):232-238 · October 2007 DOI: 10.1016/j.micromeso.2007.03.036).

“Sol-Gel Science: The Physics and Chemistry of Sol-Gel Processing” by C. Jeffrey Brinker and George W. Scherer will help with better understanding and explanation of the processes in the liquid-sol-gel system and crystallites formation.

11.   L. 114 and Fig. 2. The method is called X-ray diffraction or XRD, isn’t it? Please, analyze carefully the pattern. Even if it is poorlycrystalline / semicrystalline, the characteristic lines must appear. Please compare your patterns on fig. 2 and fig. 3. Are you sure it is the same phase?

12.   L. 120 and Fig.3                Is it an XRD spectrum of your hydrothermally treated hydrogel? And that spectrum confirms the expected phase crystallization? Provide the analysis, please.

13.   L. 126-130           it is not clear how the treatment at 25°C, 100 °C was done. What is the difference between treatment at 100 °C and drying at 100 °C?

14.   L. 129    omit coma before the colon.

15.   L. 145     rewrite sentence

16.   Title of Fig. 4      it seems, that "curing" is not an appropriate term here. Maybe you meant treatment conditions? Please correct it all through the manuscript.

17.   Analysis of XRD spectrum Fig. 4                Grammar must be checked and the paragraph must be rewritten. Moreover, it is clear that you have more than one phase formation. Take the articles of Kostov-Kytin, Anderson, Cliefrield, and Kuznicki, compare their patters with yours. You`ll see there is no such thing as gel`s peak. You need to assay that peak to the proper phase that formed in your gel. By the way, did you rinse your gels from salts? If no, why don’t you consider the probability of salt phase presence?

18.   L. 164    you meant Figure 5?

19.    The same line             It is a diagram

20.   Lines 172-175     you already describe the method, so no need to repeat it here.

21.   Please compare Fig. 4 and Fig. 7. Peaks shifted, Intense peak at 32 2theta disappears and wide peak at 15 2theta do not belongs to Congo Red. If you found the literature that proves and explain it? Please cite it.

22.   Lines 194-200. Did you rinse your gels after Hydrothermal treatment? Was there any liquid that separates from the gel?

23.   What is the pH of your gels before and after you add Congo Red?

24.   Line 212                “respectively” goes with coma and it follows the data. Please revise the sentence.

25.   Conclusions        L.236-238            not clear the idea.

26.   L.241      32 2theta is not a characteristic peak of ETS-10 phase, isn’t it?

Author Response

ANSWER TO REFEREE N° 1

Dear Referee,

We thank you for your valuable suggestions and for giving us the opportunity to make our work clearer and more correct. We have cut and rewritten some parts thanks to the referees who highlighted some critical points. We also feel to thank you for the kind and delicate way in which you proposed your suggestions. Here are the answers to your suggestions.

Best regards

Authors

Major corrections:

- Grammar, punctuation, and style must be seriously revised.

The manuscript has been revisited in English

- The methodology part needs revision. Synthesis details have to be omitted from the introduction and discussion. At the same time, the procedure of synthesis must be presented clearer and more detailed.

In the introduction following sentences were removed “The pellets were heat treated, after their forming”, “The pH indicator did not influence the system reactivity and its catalytic properties”. In the paragraph Material and Methods was added following sentence “"In particular, the following weight amounts of the reactants were used to prepare the hydrogel:  Sodium silicate solution = 33.16 g; Sodium hydroxide solution = 9.23 g; potassium fluoride solution: 8.71 g; hydrochloric acid = 4.70; titanium tetrachloride solution = 7.65 g; distilled water = 32.60 g. " (line 94-97).

- XRD patterns not properly analyzed, literature references need to attributed to peaks on patterns.

The following sentence was added at line 136:”The following Figure 3 shows the XRD spectrum of the ETS-10 crystalline phase obtained. The XRD patterns are in good agreement with the published ETS-10 data [22, 24].”

- Rewrite the Conclusions.

Conclusions were rewritten: “The method of preparation of self-bonded pellets is used for the preparation of pH-responsive ETS-10 phase crystalline pellets. This method does not use as binder substances different from the system. It allows us to hold the chemical characteristics typical of the ETS-10 phase. The best system identified that allows to have pH-responsive ETS-10 phase crystalline pellets, combining high crystallinity and high mechanical strength, is obtained for an ETS-10crystalline/ xerogel = 3 ratio and treatment at 100 °C.The pellets obtained with Congo Red, undergoing toning tests showed the ability to change color reversibly depending on the pH of the solution. Given the significant importance of the ETS-10 phase and its wide applicability, in this form and with these characteristics, it is proposed as an excellent material to be used in catalytic processes, for its advantageous recovery and its valuable characteristic of a simple pH-sensor for the control of the process parameters and performance”.

- Do you consider biotoxicity and cancerogenic properties of the pigment you have chosen? You started the paper with attitude to solve the Environmental problems, in such case toxic dye is not the best choice. Or the introduction must be focused on the catalysis that can be benefited by such an idea.

We thank you very much for your relevant note. Immersion tests of pellets in water have shown that they maintain their shape and do not release the pigment in water. It must however be added that the quantity used is very low. Furthermore, it must be emphasized that these materials should be used in preliminary water treatments for the removal of specific pollutants. Certainly, the possible applications in catalytic processes play an important role alongside their last possible use.

The following sentence has been added to the text. (line 227-229) “ The pellets obtained dipped in water for 2 hours at room temperature have shown to preserve their shape, their mechanical characteristics and their color, thus allowing to exclude the loss of the indicator in the water and confirming that it is well anchored inside of pellets.”

- What catalytic reaction are you going to use as a test reaction? how do you choose the size of the pellet?

In this work we wanted to research the experimental conditions to obtain ETS-10 phase pellets that have the ability to change color. Certainly in future studies, the catalytic processes that can benefit from this material will be sought.

Minor corrections:

- 1. P1, Abstract- Rewrite first sentence, it is confusing due to a dot (line 15) in the middle of the sentence.

The sentences was corrected (line 15-17)

-2. Keywords. You have mentioned in abstract catalysis as a main area of application, but say nothing about it in the Keywords. Maybe, highlighting that moment in keywords will be helpful.

catalysis” has been added to keywords.

- 3.Introduction. Line 37. 2nd sentence is too much generic. It is advisable to make it more specific to the purpose or to avoid it at all.

If this does not create problems we would like to keep this sentence.

L. 49 Provide some examples of target environmental pollutions

This sentences was added: “as polycyclic aromatic hydrocarbons, cycloalkanes, benzene, toluene, ethylbenzene..etc.” ( now line 52)

b.L. 67To our knowledge, there are ….c. L. 71other phases of Titanosilicate materials?

 We conferm: To our knowledge there are no studies on pH-responsive ETS-10 phase crystalline pellets

Lines 71-78. Those are not the introduction sentences. It would be better to move them to part 2. Materials and methods

Some sentences were removed from the introduction.

4.Materials and methods .1st sentence need to be revised: make it simpler, consequently, punctuation will be easier.

The sentence was correct in “To synthesize the ETS-10 phase pellets a method was applied reported the literature for other phases [50].( line 83-84)

5.L. 83 an initial gel

It was corrected

L. 90 Sentence need to be rewritten (For example: Thereafter, hydrogel was dried at…. and obtained xerogel was grinned using such and such tool). Work on synonyms, there is too much repetition. Working with gels you ought to know that fresh gel you obtained is a hydrogel. It is important to mention because you are drying them afterward obtaining xerogels. Please, use scientific terminology.

The entire manuscript and figures have been corrected using the appropriate words  “Hydrogel” and .“xerogel”

Figure 1.a It is not informative and, rather, confusing. It is not clear whether it is a white paste or white gel. When you are taking gel out of the beaker, are the portions of gel keeping their shape? If you want to prove and show that it really was gel your pictures must be different. For example, the stirring stick must stand upright, not touching the side of the beaker. Another way: you incline the beaker, but gel’s surface does not move. Picture of the gel without the beaker, that holding its own shape would be more informative as well.

Actually what you say is right, but for our experience that we have been working on the ETS-10 phase for years, its gel never presents a particular consistency. The photo represents the real state of the obtained gel and in the literature it is indicated as gel.

8 Figure 1  it would be nice to present a picture of dried gel before it was grined as well.

I currently have no other photos

9.Lines 103 and 106 do you mean crystals or crystallites?

crystals

Lines 100-106     It is good that you are trying to explain and not just providing the facts. Conditions for crystallization the titanosilicate phases (pH, composition and temperature influence) were studied in details by V. Kostov-Kytin and colleagues (Hydrothermal synthesis of microporous titanosilicates.  Microporous and Mesoporous Materials 105(3):232-238 · October 2007 DOI: 10.1016/j.micromeso.2007.03.036). “Sol-Gel Science: The Physics and Chemistry of Sol-Gel Processing” by C. Jeffrey Brinker and George W. Scherer will help with better understanding and explanation of the processes in the liquid-sol-gel system and crystallites formation.

We have added the following sentence in line 125: Kostov-Kytin et al.[51] have also analyzed the conditions for crystallization of titanosilicate phases, however these authors have obtained pure crystals and not self-bonded pellets.

L. 114 and Fig. 2. The method is called X-ray diffraction or XRD, isn’t it?

The expression was corrected.

Please, analyze carefully the pattern. Even if it is poorlycrystalline / semicrystalline, the characteristic lines must appear. Please compare your patterns on fig. 2 and fig. 3. Are you sure it is the same phase?

Figure 2 shows the spectrum of the xerogel, then in an amorphous phase and where the first aggregate formations are present, Figure 3 shows the spectrum of the ETS-10 crystalline phase. Therefore the two spectra cannot have the same peaks.

L. 120 and Fig.3 Is it an XRD spectrum of your hydrothermally treated hydrogel? And that spectrum confirms the expected phase crystallization? Provide the analysis, please.

Figure 3 is the XRD spectrum of the crystalline ETS-10 phase. This spectrum was inserted to compare the XRD spectra of the pellets and to confirm the presence of the ETS-10 phase within the pellets. The analyzes are provided through figures 5.

L. 126-130           it is not clear how the treatment at 25°C, 100 °C was done. What is the difference between treatment at 100 °C and drying at 100 °C?

The description of the treatment has been improved in the text ( line 152-158):

In order to find the best preparation conditions, the pellets were subjected to three distinct types of treatment: (1) they were left at room temperature (25 °C)  for a day; (2) they were placed directly in the oven at 100 °C for a day; .(3) they were subjected to hydrothermal synthesis at 190°C for three days. In the latter case, the pellets were placed in Morey type autoclaves in which a support made of Teflon is positioned which has the purpose of maintaining the raised pellet at 5 mm from the bottom. At the end of the  hydrothermal reaction, the pellets were removed from the autoclave, washed with distilled water and dried in an oven at 100 ° C for one day.

L. 129    omit coma before the colon.

The comma has been omitted

L. 145     rewrite sentence

The sentence was rewritten

Title of Fig. 4      it seems, that "curing" is not an appropriate term here. Maybe you meant treatment conditions? Please correct it all through the manuscript.

The expression has been corrected in the manuscript

Analysis of XRD spectrum Fig. 4  Grammar must be checked and the paragraph must be rewritten. Moreover, it is clear that you have more than one phase formation. Take the articles of Kostov-Kytin, Anderson, Cliefrield, and Kuznicki, compare their patters with yours. You`ll see there is no such thing as gel`s peak. You need to assay that peak to the proper phase that formed in your gel. By the way, did you rinse your gels from salts? If no, why don’t you consider the probability of salt phase presence?

The spectra shown in Figure 5 are those of the pellets. In a highly crystalline phase the gel peak (32 2 theta) will not be present, but in our case the pellets contain the gel that was added as a binder and therefore the presence of the peak 32. 2 theta is present until the gel does not crystallize completely in ETS-10. Most probably the gel peak at 32 2 theta is to be attributed to titanates. The gel that is obtained cannot be washed because otherwise there is the loss of the nutrients that are necessary for the growth of the ETS-10 phase crystals.

L. 164    you meant Figure5?

The error was corrected

The same line             It is a diagram

it has been corrected

Lines 172-175     you already describe the method, so no need to repeat it here.

The description was inserted to identify the pellets.

Please compare Fig. 4 and Fig. 7. Peaks shifted, Intense peak at 32 2thetadisappears and wide peak at 15 2theta do not belongs to Congo Red. If you found the literature that proves and explain it? Please cite it.

Rightly our statement is not substantiated. We proceeded to remove the sentence.

Lines 194-200. Did you rinse your gels after Hydrothermal treatment? Was there any liquid that separates from the gel?

The classical method of preparation of the ETS-10 precursor gel does not include a wash. Any washing of it would lead to the loss of many nutrients needed for crystal growth. In any case, no liquid separates from the gel

What is the pH of your gels before and after you add Congo Red?

The pH of the ETS-10 phase precursor gel is around 10.5. The addition of Congo Red has not brought any important change in pH, probably due also to the small amounts of it used.  In any case the following sentence has been added in the text: "the pH of the hydrogel is 10.5. (now line 99); “The addition of the congo red did not lead to a change in the pH of the system that remains around 10.5”. (now line 166)

Line 212                “respectively” goes with coma and it follows the data. Please revise the sentence.

It was corrected

Conclusions        L.236-238            not clear the idea.

The conclusion were rewritten

L.241      32 2thetais not a characteristic peak of ETS-10 phase, isn’t it? 2theta is the most intense peak of xerogel but not of the ETS-10 crystalline phase

Reviewer 2 Report

This is a decent piece of small work by De Luca et al. The synthesis are well designed and presented with sufficient enough detail to warrant reproducibility. The pH responsiveness results from a classic indicator Congo Red. There’s only a few questions that I would like to draw the authors’ attention to before recommending it for publication. 

Please test larger pH range, not only pH 1 and 13. 

The thermal degradation of the dye should be investigated since the crystallization process uses high temperature, which could potentially decompose the dye. 

Does the prepared gel swell in water or any other kind of solvent? 

Author Response

ANSWERS TO REFEREE 2

Dear Referee,

 We thank you for your valuable suggestions and for giving us the opportunity to make our work clearer and more correct. Here are the answers to your suggestions.

Best Regards

Authors

-Please test larger pH range, not only pH 1 and 13.

These two pH values were chosen to return to extreme conditions of acidity and basicity so as to also involve intermediate values.

-The thermal degradation of the dye should be investigated since the crystallization process uses high temperature, which could potentially decompose the dye. 

We thank you for the relevant note. The pellets that proved adequate were those treated at 100 °C. These have shown the ability to change which leads us to state that the indicator has returned well to the treatment temperature. Certainly in future work this aspect will also be deepened through NMR analysis.

-Does the prepared gel swell in water or any other kind of solvent? 

The gel we prepared for the synthesis of the ETS-10 phase uses water as solvent, as also reported in other previous works (see references).

Reviewer 3 Report

There are several flaws with this study:

1) The experimental section is not well described. The authors should provide the correct amount of the reactants used in the synthesis. The experimental description of the syntheses will not allow a straightforward reproducible repetions  of the experiments;

2) The authors do not make it clear the reasons for  using ETS-10. Why  have they  used this specific titanosilicate? and not other aluminosilicate such as faujasite zeolite ZSM-5 or zeolite Beta?  

3) The graphical quality of the XRD patterns are of very low quality. Furthermore there are no BET study and SEM data. 

This study will not cause a major impact in the field of environmental remediation. 

Author Response

ANSWERS TO REFEREE 3

Dear Referee,

We thank you for your valuable suggestions and for giving us the opportunity to make our work clearer and more correct. Here are the answers to your suggestions.

Best regards

Authors

1) The experimental section is not well described. The authors should provide the correct amount of the reactants used in the synthesis. The experimental description of the syntheses will not allow a straightforward reproducible repetions of the experiments;

The following sentence was inserted in the manuscript.

"In particular, the following weight amounts of the reactants were used to prepare the hydrogel: Sodium silicate solution = 33.16 g; Sodium hydroxide solution = 9.23 g; potassium fluoride solution: 8.71 g; hydrochloric acid = 4.70; titanium tetrachloride solution = 7.65 g; distilled water = 32.60 g. " (Line. 94-97)

2) The authors do not make it clear the reasons for using ETS-10. Why have they used this specific titanosilicate? and not other aluminosilicate such as faujasite zeolite ZSM-5 or zeolite Beta?  

The ETS-10 phase has been studied to deepen our studies that our group has been performing on this material for years. Moreover in many studies the ETS-10 phase has shown itself to be highly valid in ion exchange and catalysis processes, as reported in the references and at lines 47-52. Rightly, as you suggest, this may be applied in the future for other equally important phases.

3) The graphical quality of the XRD patterns are of very low quality. Furthermore there are no BET study and SEM data. 

Image quality has been improved. At the moment we are not able to provide BET and SEM analysis, but certainly future work will take this characterization into consideration.

This study will not cause a major impact in the field of environmental remediation.

The advantage of this material is that with these characteristics, it can be used mainly in catalytic processes, for its advantageous recovery and its valuable characteristic of a simple pH-sensor for the control of the process parameters and performance.

Round 2

Reviewer 3 Report

The authors have improved the quality of the manuscript compared to the previous version. Therefore I recommend the publications of the manuscript. 

This manuscript is a resubmission of an earlier submission. The following is a list of the peer review reports and author responses from that submission.